# Neurofibromin Structure, Functions and Regulation

**DOI:** 10.3390/cells9112365

**Published:** 2020-10-27

**Authors:** Mohammed Bergoug, Michel Doudeau, Fabienne Godin, Christine Mosrin, Béatrice Vallée, Hélène Bénédetti

**Affiliations:** Centre de Biophysique Moléculaire, CNRS, UPR 4301, University of Orléans and INSERM, CEDEX 2, 45071 Orléans, France; bergoug_mohammed@yahoo.fr (M.B.); michel.doudeau@cnrs-orleans.fr (M.D.); fabienne.godin@cnrs-orleans.fr (F.G.); christine.mosrin@cnrs-orleans.fr (C.M.); beatrice.vallee@cnrs-orleans.fr (B.V.)

**Keywords:** neurofibromin, structure, function, localization, interactions, post-translational modifications

## Abstract

Neurofibromin is a large and multifunctional protein encoded by the tumor suppressor gene *NF1*, mutations of which cause the tumor predisposition syndrome neurofibromatosis type 1 (NF1). Over the last three decades, studies of neurofibromin structure, interacting partners, and functions have shown that it is involved in several cell signaling pathways, including the Ras/MAPK, Akt/mTOR, ROCK/LIMK/cofilin, and cAMP/PKA pathways, and regulates many fundamental cellular processes, such as proliferation and migration, cytoskeletal dynamics, neurite outgrowth, dendritic-spine density, and dopamine levels. The crystallographic structure has been resolved for two of its functional domains, GRD (GAP-related (GTPase-activating protein) domain) and SecPH, and its post-translational modifications studied, showing it to be localized to several cell compartments. These findings have been of particular interest in the identification of many therapeutic targets and in the proposal of various therapeutic strategies to treat the symptoms of NF1. In this review, we provide an overview of the literature on neurofibromin structure, function, interactions, and regulation and highlight the relationships between them.

## 1. Introduction

Neurofibromatosis type 1 is an autosomal dominant disorder caused by inherited or de novo germline mutations in the *NF1* tumor suppressor gene [1,2]. It is the most common tumor-predisposing disease in humans. It affects approximately one in 3000 live births and patients present widely heterogeneous clinical manifestations, even within the same family. Hallmark traits of the disease include pigmentary lesions and various types of peripheral nervous system tumors, cutaneous neurofibromas (cNFs), plexiform neurofibromas (pNFs), and malignant peripheral nerve sheath tumors (MPNSTs). However, some individuals develop other symptoms, such as skeletal abnormalities, brain tumors, learning disabilities, attention deficit, and social and behavioral problems. *NF1* has been shown to be an essential gene for embryonic development and mice lacking a functional gene die in utero from cardiovascular defects [3]. To date, more than 3000 different germline mutations within the *NF1* gene have been identified and shown to be pathogenic [4,5], but acquired somatic mutations in *NF1* are also found in a wide variety of malignant neoplasms unrelated to NF1 [6]. *NF1* encodes neurofibromin, a large multifunctional protein that is ubiquitously expressed but of which the highest levels are found in the neuronal cells of adults. Allelic variation, second-hit events in the *NF1* gene, germline-specific genetic context, and epigenetic changes, as well as tissue-specific functions of neurofibromin may account for the profound degree of clinical heterogeneity in NF1 [7,8]. Detailed studies of the molecular and cellular properties of neurofibromin are required for a full understanding of the diverse phenotypes associated with the disease and their progression, as well as for the identification of new therapeutic targets to develop pharmacological therapies against NF1. This review summarizes three decades of research on neurofibromin by focusing on its structure and molecular aspects of its functions and regulation.

## 2. The *NF1* Gene

Neurofibromatosis type 1 is caused by mutations within the *NF1* gene. Early genetic linkage analysis located the *NF1* gene near the centromere on the long arm of chromosome 17 [9]. Subsequent studies that showed NF1 patients to have translocations *t*(1;17) and *t*(17;22) [2] and deletions and point mutations [1,10] led to mapping of the *NF1* gene to the 17q11.2 locus. *NF1* is a large gene that spans 350 kb of genomic DNA sequence [1]. Its transcript is 11 to 13 kb long, with an 8454-bp open reading frame [11] and 3.5 kb of 3’ untranslated region [12]. It contains 60 exons and is ubiquitously expressed [2] (Figure 1).

Three active genes lie within intron 35 (27b in the previous numbering) of *NF1* and are transcribed from the opposite strand: *EVI2A* (ecotropic viral integration site) [13], *EVI2B* [14], and *OMGP* (oligodendrocyte myelin glycoprotein) [15] (Figure 1). *EVI2A* and *EVI2B* are the human homologues of the murine *Evi-2A* and *Evi-2B* genes, which encode proteins involved in retrovirus-induced murine myeloid leukemogenesis [13,14]. *OMGP* encodes a glycoprotein specifically expressed in oligodendrocytes [16]. None of these genes have been found to be mutated in NF1 patients [13,14,15].

The presence of multiple tumors in affected patients and the identification of somatic mutations within the *NF1* gene in sporadic tumors independently of NF1 disease has led to *NF1* being designated as a tumor suppressor gene [17].

## 3. Neurofibromin Protein

The *NF1* gene encodes a large protein of 2818 amino acids called neurofibromin [11]. Sequence analysis has shown homology between *NF1* and the *Saccharomyces cerevisiae IRA1* and *IRA2* genes, which negatively regulate the RAS-cyclic AMP pathway [18]. In addition, a 360 amino-acid portion of neurofibromin appears to be homologous to the catalytic domains of *IRA* gene products and mammalian p120GAP (GTPase-activating protein), suggesting that neurofibromin possesses a Ras-GAP function [19]. This was confirmed by the capacity of this portion of the *NF1* gene to complement the function of the *IRA1* and *IRA2* genes and restore a wildtype phenotype when expressed in *IRA*-mutated yeast by stimulating the intrinsic GTPase activity of the yeast Ras protein or the GTPase activity of human Ras expressed in yeast. These studies established neurofibromin as a GTPase-activating protein for Ras (Ras-GAP). The neurofibromin domain carrying the GAP function was named GRD for GAP-related domain [18,20,21,22,23]. The development of specific antibodies against the *NF1* gene product allowed the identification of neurofibromin as a protein with a molecular mass of 250 to 280 kDa that is ubiquitously expressed, but which exhibits its highest levels within the nervous system in neurons, non-myelinating Schwann cells, and oligodendrocytes, as well as in the adrenal medulla, leukocytes, and testis [24,25,26].

In addition to yeast Ira1 and Ira2 proteins, neurofibromin homologous proteins appear to be present in a wide range of eukaryotic lineages ranking from fungi to mammals with more or less sequence identity [27]. For instance, *Drosophila* neurofibromin and mouse neurofibromin are, respectively, 60% [28] and 98.5% identical to human neurofibromin [29]. These two organisms have been used as models to study neurofibromin functions and led to many discoveries that will be discussed later in this review. More recently, a novel porcine model of NF1 was developed. This new model displays many phenotypes of the human disease and therefore provides a valuable tool to study NF1 disease and evaluate new therapies [30].

Sequence analysis showed that neurofibromin is highly conserved in mammals. Amino acid sequence of human neurofibromin is 98.5% and 99.4%, respectively identical to its counterpart in rat and dog [31]. In primates, neurofibromin amino acid sequence remains almost unchanged [28]. Human neurofibromin is 100% identical to its orthologues in chimpanzee (*Pan troglodytes*) [31], in gorilla (*Gorilla gorilla*), in orangutan (*Pongo abelii*), in gibbon (*Nomascus leucogenys*), and in marmoset (*Callithrix jacchus*) [28], 96% in rhesus macaque (*Macaca mulatta*), and 83% in lemuriforme mouse lemur (*Microcebus murinus*) [28]. This indicates a strong selective pressure on neurofibromin structure and function [28,31].

## 4. Neurofibromin Isoforms

Several transcript variants resulting from the alternative splicing of the pre-messenger RNA of *NF1* have been identified. RT-PCR analysis has shown tissue-specific expression and regulation of these transcript variants during development. Some are weakly expressed, whereas others show high expression levels [32]. The most abundant form of *NF1* mRNA is that containing 57 exons and encoding a 2818 amino acid neurofibromin protein [12].

Identification and functional studies of the protein products have been carried out only on a few of these transcripts. Five alternatively spliced exons, leading to the expression of different *NF1* isoforms, have been particularly studied: 9a, 10a-2, 23a, 43, and 48a (according to the previous nomenclature) (Figure 2).

### 4.1. Alternative Splicing of Exon 23a

#### 4.1.1. Type 1 and Type 2 Isoforms

Exon 23a is located within the GRD. Its alternative splicing results in two types of transcripts: one, as initially identified, encoding neurofibromin isoforms comprising 2818 aa, and the other carrying a 63-nucleotide insertion and encoding a neurofibromin isoform comprising 2839 aa, which contains 21 additional amino acids in the GRD. Nomenclature about these isoforms is very confusing in the literature and databanks since either isoform has been called isoform I or II (or neurofibromin type 1 or type 2). In the rest of the review, we will follow the nomenclature of UniProt and will call neurofibromin isoform I (or type 1), the protein of 2818 aa (Accession: NP_000258.1 in NCBI Genbank), and neurofibromin isoform II (or type 2), the protein of 2839 aa (Accession: NP_001035957.1 in NCBI Genbank). Neurofibromin type 2 is also capable of complementing *IRA1* and *IRA2* in *Saccharomyces cerevisiae* but has lower Ras-GAP activity than neurofibromin type 1. RT-PCR analysis showed that the type 2 transcript is ubiquitously expressed, similar to type 1, but with differential expression between tissues. Neurofibromin type 2 was shown to be preferentially expressed in differentiated cells [33,34]. These two isoforms have also been observed in other species (mice, chickens, rats, and cows), indicating a high level of conservation throughout evolution.

#### 4.1.2. Differences in the Function of the Two Isoforms

The generation of embryonic stem cells exclusively expressing either isoform I or II showed the Ras/ERK (extracellular signal-regulated kinase) pathway to be more active in isoform II, but the cAMP level was unchanged [35], thus confirming the observation made in yeast by Andersen et al. (1993) [33].

Neurons expressing neurofibromin type 2 have also been shown to display higher Ras/ERK signaling as well [35].

Nguyen et al. (2017) [36] generated mice expressing only isoform II in all tissues. These mice, called *NF1 23a/23a*, were viable, fertile, and showed no physical abnormalities. However, the brains of *NF1 23a/23a* mice exhibited higher Ras/ERK pathway activity than those of wildtype mice, as well as learning and memory difficulties (short-term and long-term spatial memory).

These observations suggest that the alternative splicing of exon 23a is a regulatory mechanism of neurofibromin Ras-GAP activity. The presence of exon 23a therefore reduces the Ras-GAP activity of neurofibromin and ensures appropriate activation of the Ras/ERK signaling pathway.

#### 4.1.3. Expression of the Two Isoforms

RT-PCR analysis showed that the levels of mRNA type 1 and type 2 differ between tissues, and even display variations within the same tissue during different stages of embryonic development and in adulthood (Gutman et al., 1995).

In adult rats, *NF1* mRNA is widely expressed in brain (cerebral cortex, brainstem, and cerebellum), with a predominance of isoform I. The spinal cord and testis display an equivalent level between the two isoforms. Isoform II is predominantly expressed in the adrenal glands and ovaries. Although type 1 is the main isoform found in neurons, type 2 is predominantly found in glial cells.

During embryonic development in rats, expression of the two isoforms in brain is detected at E14 and increases until E16, without any change in the predominance of isoform I. The expression of isoform I decreases in the spinal cord until the two isoforms are expressed at equivalent levels by the first postnatal week. The opposite has been observed in the testis during the first 14 postnatal days. Indeed, the expression of isoform I increases and an equivalent level between the two isoforms is reached in adults. Although *NF1* mRNA of both isoforms is detected in the kidneys, lungs, heart, and skeletal muscles during embryonic development and the first two weeks of postnatal life, very little is detected in adult rats in these tissues. These observations suggest a functional difference between the two isoforms of neurofibromin and an important role of the proteins in tissue development [37].

### 4.2. Alternative Splicing of Exon 48a

The insertion of 54 nucleotides from exon 48a into *NF1* mRNA produces neurofibromin type 3, which contains 18 additional amino acids at its C-terminus. The generation of antibodies specific to this isoform showed that neurofibromin type 3 is expressed exclusively in the heart and muscles. It is highly expressed in rat embryos and its expression decreases after the seventh day of postnatal life. Neurofibromin type 4 contains both exon 23a and 48a [38].

### 4.3. Alternative Splicing of Exon 9a

Exon 9a is an insertion of 30 nucleotides after nucleotide 1,260 of the cDNA of *NF1* and encodes 10 amino acids in the N-terminal region. Neurofibromin containing this exon is exclusively expressed in CNS neurons and is enriched in forebrain neurons. Its expression increases during the first postnatal week in mice, suggesting a role of this isoform in the maturation and differentiation of neurons [39,40].

### 4.4. Alternative Splicing of Exon 10a-2

Neurofibromin containing exon 10a-2 has 15 additional amino acids inserted in the N-terminus between exons 10a and 10b. It is ubiquitously expressed but at low levels. Sequence analysis showed that this isoform contains a transmembrane segment that is absent from the other isoforms that may account for its specific localization to perinuclear granular structures, such as the endoplasmic reticulum, suggesting that this neurofibromin isoform has a role in intracellular membranes [41].

### 4.5. Other Isoforms

Another transcript variant, *NF1 delta E43*, in which exon 43 is deleted, was identified by Vandenbrooke et al. (2002) [32]. Relative to the general expression of *NF1*, it is highly expressed in the lungs, liver, placenta, kidneys, and skeletal muscle of adult humans [32], tissues in which *NF1* expression is high during embryonic development but barely detectable in adults [37]. Exon 43 contains a functional nuclear localization signal (NLS) [42]. The particularly low expression of *NF1 delta E43* (devoid of the sequence encoding the NLS) in neurons relative to other tissues suggests that neurofibromin has an important nuclear function in neurons [42].

## 5. Neurofibromin Structure and Domains

Neurofibromin is a multidomain protein consisting of an N-terminal cysteine-serine-rich domain (CSRD), a central GAP-related domain (GRD), including a tubulin-binding domain (TBD) at its N-terminus, followed by a phospholipid- and protein-interaction domain, SecPH, and a C-terminal domain (CTD) (Figure 3). To date, only the 3D structures of the GRD and SecPH domains have been resolved, representing 25% of the entire protein. However, recently, structural data have been obtained on full-length neurofibromin using a series of biochemical and biophysical experiments indicating that full-length neurofibromin forms a high-affinity dimer [43].

### 5.1. The GAP-Related Domain (GRD)

The GRD presents similarities with the catalytic domains of other GAPs, including the Ira1 and Ira2 proteins of the yeast *Saccharomyces cerevisiae* and mammalian p120GAP [19]. Biochemical analysis showed that the neurofibromin-GRD acts as a bona-fide GAP that promotes the hydrolysis of the active form of Ras (GTP-bound Ras) into an inactive form (GDP-bound Ras) [18,19,20,21,22]. In 1998, Scheffzek et al. [44] resolved the crystallographic structure of the neurofibromin-GRD, which was obtained from a proteolytically-treated fragment spanning residues 1198 to 1530 of neurofibromin (NF1-333). It is a helical protein similar in structure to the catalytic domain of the Ras-GAP protein p120GAP (GAP334). NF1-333 is composed of two domains: (1) a central domain (NF1c) that contains residues conserved within Ras-GAP proteins, representing the minimum Ras-GAP module, and (2) an extra domain (NF1ex), consisting of residues surrounding NF1c at the N- and C-terminus of NF1-333, which displays remarkable structural similarity with GAP-334, despite the absence of sequence homology with the corresponding segment (Figure 4).

Scheffzek et al. (1998) [44] also identified the Ras-binding site as the groove in the surface of NF1c, which is bordered mainly by a finger loop (L1c) along with part of helix α2c and a variable loop (L6c) (Figure 4)**.**

Based on the structures of the neurofibromin GRD (NF1-333) [44] and the Ras-p120GAP complex [45], a structural model of the Ras-neurofibromin GRD complex was proposed [44], showing that neurofibromin-GRD provides an arginine residue (arginine finger R1276) to the active site of Ras to stabilize and orient the catalytic Ras residue, Q61, for an in-line nucleophilic attack on the gamma-phosphate of GTP.

A patient mutation substitution of this arginine to a proline residue resulted in an 8000-fold decrease in Ras-GAP activity [44,46].

More recently, the 3D structure of a complex between K-Ras (GMPPNP-bound) and a fragment of the neurofibromin-GRD (GAP255: residues 1209 to 1463), corresponding to the minimum catalytic region (the C- and N-terminal regions forming the extra domain were excluded), was resolved [47] (Figure 5), confirming the insights given by the model. This was the first structure of a Ras-neurofibromin GRD complex and of a Ras-Ras GAP complex in the ground-state conformation.

### 5.2. The SecPH Domain

Bioinformatics analysis allowed Aravind et al. (1999) [48] to predict a Sec14-like domain, homologous to the lipid-binding domain of the yeast *Saccharomyces cerevisiae* phosphatidylinositol transfer protein Sec14p [48], in neurofibromin. This domain of neurofibromin was subsequently produced, purified, and crystallized [49].

In 2006, D’Angelo et al. [50] resolved the 3D structure of the functional domain SecPH, comprised of residues 1560 to 1816 and located C-terminal to the GRD domain. SecPH appears to be a bipartite module, composed of the previously predicted Sec14-like domain in the amino-terminus (residues 1560–1698) and an unexpected protein-protein interaction module pleckstrin homology PH-like domain (residues 1715–1816). The two domains are connected by a partially helical linker peptide (residues 1699–1714) [50].

Structural analysis showed that neurofibromin-Sec is a lipid-binding cage. A central β-sheet forms the basis of its hydrophobic lipid-binding cavity. This cavity is closed by a lid-helix that blocks the ligand entry site. A β-protrusion derived from the PH-like domain interacts with the lid-helix of neurofibromin-Sec to maintain it in a closed conformation, which prevents the access of lipid ligands to the lipid-binding site within the Sec cavity (Figure 6A).

The obtained structure suggests that an open conformation would clash with the β-protrusion of the pleckstrin homologous (PH) domain (Figure 6B) [50]. Overlay assays showed that the SecPH domain binds to the membrane-immobilized phospholipid phosphorylated phosphatidylinositol (PIP). Neurofibromin-PH alone is not sufficient to bind PIP. Using lipid exchange assays and mass spectrometry, Welti et al. (2007) [52] showed that the Sec14-like portion binds to glycerophospholipids.

Based on these observations, a hypothetical model was proposed, according to which conformational changes of the PH domain regulate lipid binding in the Sec portion. Two ligands are involved in this model. Interaction with ligand A induces conformational changes in the β-protrusion of the PH domain, which then releases the lid-helix of the Sec domain and opens the lipid-binding cage, thereby allowing binding of the second ligand B (glycerophospholipids) (Figure 7) [46,50].

It is possible that Sec ligand B may be a protein and not only a lipid. Since 2006, various protein partners have been identified for Sec (LIMK2, VCP) and PH (R-5HT6). However, no structural data on the complexes are available and it is not known whether this proposed mechanism actually takes place with these or other partners. Furthermore, the physiological meaning of such a mechanism remains an enigma. However, given the proximity of the GRD domain (it is adjacent to the Sec domain in neurofibromin), it is possible that it may be involved in the fine regulation of neurofibromin Ras GAP activity.

### 5.3. The Cysteine-Serine Rich and C-Terminal Domains (CSRD and CTD)

Neurofibromin contains two other domains: a cysteine-serine-rich domain (CSRD, residues 543–909) and a C-terminal domain (CTD, residues 2260–2818) [53], which are known to harbor phosphorylation sites [54].

The CSRD is phosphorylated by both protein kinase A (PKA) and protein kinase C (PKC) [54,55]. Its PKC-dependent phosphorylation increases the Ras-GAP activity of neurofibromin by allosteric regulation and enhances its association with actin [55].

The CTD contains a functional nuclear localization signal [42] and is also phosphorylated by PKA. CTD phosphorylation on serines 2576, 2578, 2580, and 2813 and threonine 2556 is required for its interaction with 14-3-3 protein, which negatively regulates the Ras-GAP activity of neurofibromin [56,57]. The CTD is responsible for the role of neurofibromin in the regulation of the metaphase to anaphase transition [23] and acts as a tubulin-binding domain [58]. It has been shown to be phosphorylated on Ser 2808 by PKC-ε, which is important for neurofibromin nuclear localization [58]. The CTD also interacts with other neurofibromin partners, such as CRMP2 [59], syndecans [60], focal adhesion kinase (FAK) [61], and calcium/calmodulin-dependent serine protein kinase (CASK) [62].

### 5.4. Structural Data on Full-Length Neurofibromin

Size-exclusion chromatography multi-angle light scattering (SEC-MALS), small-angle X-ray and neutron scattering (SAXS and SANS, respectively), and analytical ultracentrifugation performed on purified full-length human neurofibromin produced in baculovirus-infected insect cells, demonstrated that full-length neurofibromin forms a high-affinity dimer [43]. This dimerization was shown to take place in vivo in human cells and was visualized as a pseudo-symmetric dimeric particle after transmission electron microscopy analysis of projection images of negatively stained neurofibromin. Domains mediating neurofibromin dimerization were shown to be the TBD (tubulin-binding domain), which consists of three predicted amphipathic alpha-helices and a region localized between PH and CTD domains that contains a series of 12 predicted HEAT-like repeats commonly involved in protein-protein interactions, and often forming a complex solenoid structure incorporating a series of alpha-helices [43]. The authors emphasize that this kind of interaction has already been described with high affinity mTOR dimers. Indeed, mTOR dimerization comes from the hydrophobic interaction of a HEAT-like solenoid in the N-terminal region of a first mTOR monomer with a set of three amphipathic helices in the core domain of a second mTOR monomer [63].

## 6. Localization

### 6.1. Association with Cytoskeletal Structures

Neurofibromin was found to be a tubulin-binding protein in co-purification experiments. The domain for the interaction of neurofibromin with tubulin was first localized to the N-terminal 80 residues of the GRD [64]. This region was thus named the tubulin-binding domain (TBD) (Figure 3). The neurofibromin-tubulin interaction was shown to decrease neurofibromin GAP activity [64]. These observations were consolidated by subsequent studies using double-indirect immunofluorescent labeling and demonstrated that neurofibromin associates with cytoplasmic microtubules [65]. The interacting region was again mapped to the GRD [65,66]. More recently, Koliou et al. (2016) [58] showed that neurofibromin localizes to the mitotic spindle during mitosis through interactions with α and β tubulin. It also colocalizes to the centrosome, both during mitosis and interphase, through interaction with the centrosomal organizer γ-tubulin. These interactions were shown to be mediated by the neurofibromin-CTD, thus identifying another tubulin-binding domain in addition to the previously described TBD.

Li et al. (2001) [67] reported that neurofibromin colocalizes either with the actin or microtubule cytoskeleton in a developmentally regulated manner. In differentiating neurons, neurofibromin colocalizes with F-actin in growth cones and filopodia. In differentiated neurons, however, the association of F-actin is significantly lost in favor of tubulin colocalization, particularly at the centrosomal region [67]. In response to growth factors, PKC-mediated neurofibromin phosphorylation of the CSRD was shown to increase the association of neurofibromin with actin [55].

In 2000, Koivunen et al. [68] showed that neurofibromin associates with intermediate filaments in differentiating keratinocytes. This association was shown to be limited to the period of desmosome and hemi-desmosome formation [68].

### 6.2. Nuclear Localization

Neurofibromin was long thought to be only a cytoplasmic protein. The nuclear presence of neurofibromin in keratinocytes was an intriguing observation and suggested that neurofibromin may have other functions aside from its Ras-GAP activity [68]. In 2001, Li et al. [67] reported nuclear localization of neurofibromin in telencephalic neurons and suggested that a bipartite nuclear targeting sequence identified bioinformatically between residues 2555 and 2572 of neurofibromin (namely: KRQEMESGITTP PKMRRV, in which the underlined residues form the bipartite NLS) may be functional [67]. In 2004, Vandenbroucke et al. [42] confirmed this hypothesis and demonstrated that this in silico-predicted NLS, present within the alternatively spliced exon 43a, was functional and necessary for neurofibromin nuclear import [42].

Since then, the nuclear presence of neurofibromin has been observed in several cell types. For example, neurofibromin was identified in large-scale studies carried out in Hela cells as a phosphorylated protein of the nucleus [69] and mitotic spindle [70]. Then, Leondaritis et al. (2009) [71] showed that a fraction of neurofibromin shuttles from the nucleus to the cytoplasm of neuroblastoma cells (SH-SY5Y) upon their differentiation and the sustained PKC/Ras/ERK pathway activation mediated by phorbol ester 12-O-tetradecanoyl-phorbol-13-acetate (TPA) treatment. This may be mediated by a specific phosphorylation event that occurs in the neurofibromin C-terminal domain under these conditions [71]. Kweh et al. (2009) [61] also showed that, in addition to cytoplasmic and perinuclear colocalization, a fraction of neurofibromin colocalizes with FAK in the nucleus of normal human Schwann cells and breast cancer cells. In addition, we previously showed that a fraction of neurofibromin colocalizes with promyelocytic leukemia (PML) nuclear bodies (PML-NBs) in the nucleus of astrocytoma cells (CCF) [72]. Finally, the distribution of neurofibromin in cycling SF268 glioblastoma cells was shown to be cell-cycle-dependent. Neurofibromin, predominantly extra-nuclear at the G1/S transition, progressively accumulated in the nucleus throughout the S phase, and then became primarily nuclear prior to mitosis, gradually declining by the next G1. PKC-ε-mediated neurofibromin phosphorylation at S2808 increased the nuclear accumulation of the protein. The neurofibromin nuclear pool was preferentially localized to the nuclear matrix due to its interaction with nuclear intermediate filaments lamins A/C [58].

### 6.3. Plasma Membrane Localization

Stowe et al. (2012) [73] reported an interaction between neurofibromin and Spred1 (Sprouty-Related EVH1 domain-containing protein), a protein that has been shown to negatively regulate Ras/MAPK signaling and mutations of which cause Legius syndrome, a RASopathy that shares mild phenotypes with NF1 [74].

Spred1 is recruited to the cell membrane via interactions of its C-terminal SPR (Sprouty-Related) domain with phospholipids and caveolin-1, a protein located in the plasma membrane [75]. In turn, Spred1 recruits and translocates neurofibromin to the cell membrane via an interaction between Spred1-EVH1 (ENA-VASP Homology domain) and neurofibromin-GRD via the GAPex domain of neurofibromin [76]. Neurofibromin translocation to the cell membrane enables its localization in proximity to Ras for possibly more efficient downregulation of Ras by its Ras-GAP activity. Recently, Yan et al. (2020) [77] resolved the structure of a trimeric complex between the neurofibromin GRD, the Spred1-EVH1 domain, and K-Ras, allowing a precise analysis of the neurofibromin/Spred1 interface. This provided a rationale for the mutations observed in Legius syndrome and revealed mechanistic insights concerning K-Ras regulation.

Neurofibromin may also directly bind to caveolin-1 (Cav-1), as it was shown to be present in Cav-1-enriched membranes (CEMs) [78], suggesting that this interaction may directly target neurofibromin to the plasma membrane. Neurofibromin interactions with various members of the syndecan transmembrane protein family [60] may also play a role in its plasma membrane localization.

### 6.4. Other Reported Localization

Neurofibromin has also been reported to localize to other cellular organelles, such as the endoplasmic reticulum [79], mitochondria [80], and melanosomes [81].

## 7. Neurofibromin Functions

### 7.1. Ras-GAP Activity

Neurofibromin is a GTPase-activating protein of Ras (Ras-GAP). It downregulates the Ras signaling pathway by promoting the hydrolysis of the active form of Ras (GTP-bound Ras) to an inactive form of Ras (GDP-bound Ras) by increasing the intrinsic GTPase activity of Ras [20,21,22]. Ras was shown to be constitutively active in malignant tumor cell lines derived from NF1 patients, even though Ras and p120GAP were functionally wildtype, suggesting that neurofibromin is the main negative regulator of Ras in the tested tissues [82,83]. This may be explained by the fact that neurofibromin binds more efficiently to Ras than p120GAP [84].

#### 7.1.1. Role in Cell Growth

GRD expression normalizes active Ras levels and restores normal growth of *NF1*-deficient cell lines derived from NF1 patients [83,85]. A reduction (*NF1*^+/−^) or loss (*NF1*^−/−^) of neurofibromin expression in neural stem cells (NSCs) confers survival and a proliferative advantage as a consequence of hyperactivation of the Ras signaling pathway that was rescued by GRD expression [86].

The duration of Ras signaling is critical for signaling decisions [87]. Stimulation by different growth factors can induce different durations of Ras activation and lead to different cell responses [87]. Cichowski et al. (2003) [88] showed that neurofibromin regulates the amplitude and duration of Ras signaling pathway activation in growth-factor responses. Indeed, *NF1*-deficient mouse embryonic fibroblasts (MEFs) have been shown to be more sensitive to growth factors than wildtype MEFs. They only require non-mitogenic levels of growth factors for maximal Ras activation and proliferation [88]. Consistent with this observation, *NF1*-deficient (*NF1*^−/−^) hematopoietic cells show a hyperactive Ras signaling pathway and a high number of colonies in cultures supplemented with low concentrations of stimulating factors [89]. Henning et al. (2016) [90] described a feedback mechanism leading to stimulation of neurofibromin GAP activity to restrict the duration of growth factor-mediated Ras activation. In this study, knockdown of neurofibromin prolonged Ras-GTP accumulation in cells stimulated by epidermal growth factor (EGF) [90].

The Ras isoform preferentially inhibited by neurofibromin may differ from one cell type to another. In astrocytes, neurofibromin loss results in selective hyperactivation of K-Ras. Activation of K-Ras, but not H-Ras, is the cause of the proliferative advantage in *NF1*^−/−^ astrocytes [91]. This study provided evidence that K-Ras is the primary target of neurofibromin Ras-GAP activity and growth control in astrocytes. The authors found these observations to have important pharmacological and therapeutic implications. They suggested that molecular therapies targeting K-Ras may be the most appropriate choice in the treatment of NF1-associated tumors, in which K-Ras is specifically hyperactivated. Knowing that all Ras isoforms are farnesylated by farnesyltransferases, and that K-Ras is the only geranylgeranylated isoform, the use of geranylgeranyl transferase inhibitors alone or in combination with farnesyltransferase inhibitors (FTI) was suggested [91]. Unfortunately, a phase II trial using tipifarnib, a farnesyltransferase inhibitor, demonstrated no influence on time to progression of plexiform neurofibroma [92].

#### 7.1.2. Role in Learning

Ras inhibition by neurofibromin has been shown to be involved in learning. Costa et al. (2002) [93] showed that learning deficits and impaired long-term potentiation LTP (LTP is a synaptic plasticity mechanism involved in learning and memory), seen in *NF1*^+/−^ mice, were caused by Ras hyperactivation.

Genetic or pharmacological manipulation that decrease Ras/MAPK activity rescued these phenotypes, suggesting that Ras/MAPK inhibition could be a therapeutic strategy against learning deficits in NF1 patients [93,94,95,96]. Along this line, different clinical trials were performed on cognitive outcomes of NF1 children using statins medications (lovastatin and simvastatin, which inhibit Ras farnesylation) with no improvement in visuospatial learning or attention [97,98,99].

A further study in this context shed light on the molecular mechanisms involved in the learning disabilities in NF1 mouse models with heterozygous Cre-mediated deletion of *NF1* (resulting in hyperactivation of the ERK signaling pathway) in different key cell types of the brain (astrocytes, pyramidal neurons, excitatory and inhibitory neurons, GABAergic neurons) [100]. An increase in gamma-aminobutyric acid (GABA) release was shown to take place in inhibitory neurons of the hippocampus, resulting from the ERK-dependent phosphorylation of synapsin I. Such an increase in GABA levels in the hippocampus affected hippocampal synaptic plasticity, LTP, and learning. Indeed, a GABA antagonist rescued learning deficits associated with *NF1* deletion in inhibitory neurons [100] (Figure 8).

Oliveira and Yasuda (2014) [87] also reported that neurofibromin regulates synaptic plasticity via its Ras-GAP activity. In this study, the authors showed that neurofibromin is the major Ras inactivator in dendritic spines of pyramidal neurons of the hippocampus and that reduction of Ras inactivation in neurons expressing low levels of neurofibromin leads to impaired structural plasticity and spine loss. The observed phenotypes were rescued by the expression of the neurofibromin-GRD [87].

### 7.2. cAMP Regulation

In most mammalian cells, the cAMP-dependent protein kinase A pathway promotes growth arrest and cell differentiation [101]. In addition, cAMP disruption in NF1 mouse models has been shown to be sufficient to promote glioma formation, demonstrating the role of cAMP in the formation of certain tumors [102].

#### 7.2.1. Neurofibromin is a Positive Regulator of cAMP Levels in Various Cell Types in Both a Ras-Dependent and Ras-Independent Manner

In *Drosophila*, neurofibromin is required to promote G protein-mediated adenylyl cyclase activation. Neurofibromin-null mutant flies have low cAMP levels [103]. They are characterized by small size and learning defects that are rescued by the expression of constitutively active PKA, the downstream effector of cAMP. Pharmacological enhancement of cAMP signaling also restores learning defects in *NF1*-deficient zebrafish [104].

However, in *Drosophila*, it is clear that these phenotypes are not rescued by attenuating Ras activity [105,106,107].

Similarly, GRD expression in *NF1*-deficient murine neocortical astrocytes only partially restores neurofibromin-mediated cAMP production in response to PACAP (pituitary adenylate cyclase activating peptide), suggesting that one or more other domains outside of the GRD are involved in Ras-independent neurofibromin regulation of cAMP levels [108] into these cells.

Furthermore, elevating cAMP levels with the adenylate cyclase (AC) activator forskolin or treatment with the phosphodiesterase 4 inhibitor rolipram, but not by MEK or PI3K inhibition, have been shown to reverse the decrease in neurite length, growth cone area, and the increase of apoptosis caused by impaired neurofibromin-mediated cAMP production in *NF1*-deficient mouse embryonic brain [109]. The same results have been obtained in *NF1*-heterozygous (*NF1*^+/−^) CNS, hippocampal, and retinal ganglion cell (RGC) neurons [110,111].

#### 7.2.2. Detailed Molecular Mechanisms of Neurofibromin-Mediated cAMP Regulation

Further studies in *Drosophila* brain have shown that neurofibromin regulates cAMP by activating two distinct adenylate cyclase (AC) pathways. The first is a growth factor-stimulated neurofibromin/Ras-dependent AC pathway downstream of tyrosine kinase growth factor receptors (EGF, transforming growth factor TGF, etc.). This pathway requires Ras-GAP activity of the neurofibromin-GRD domain and Ras to stimulate AC activity, independently of Gαs. The second is a neurofibromin/Gαs-dependent pathway that is stimulated by neurotransmitters, such as serotonin and histamine. This pathway requires the CTD of neurofibromin [112]. The existence of this Ras-independent pathway is supported by a recent study showing that, by interacting with the serotonin receptor 5-HT6^R^ via its PH domain, neurofibromin promotes 5-HT6^R^ constitutive activation of the Gαs/AC pathway in striatal neurons. Both *NF1* silencing and NF1 patient mutations within the PH domain that disrupt the neurofibromin/5-HT6^R^ interaction inhibit constitutive receptor activity on the Gαs/AC pathway and reduce basal cAMP levels [113] (Figure 9).

Anastasaki and Gutmann (2014) [114] established a mechanism involving neurofibromin-dependent cAMP regulation in a Ras-dependent manner in hippocampal neurons. This process does not involve classical Ras-mediated MEK/AKT signaling but requires activation of an atypical PKC zeta by Ras, leading to GRK2-mediated Gαs inactivation [114] (Figure 9).

This pathway may represent the initial part of a more general pathway, described by Brown et al. (2012) [111], that was shown to regulate neurite growth and growth-cone formation in neurons of the CNS and involving PKA-dependent activation of the Rho/ROCK/MLC pathway (Figure 9).

We would not be exhaustive without citing the work of Lin et al. (2007) [115], who showed that neurofibromin is a binding partner of syndecan2, which induces syndecan2-dependant activation of PKA and its actin-associated downstream effectors Ena-VASP, thus inducing actin polymerization and dendritic filopodia formation and contributing to spinogenesis (Figure 9).

Unlike the examples given above, in Schwann cells, in which cAMP acts as a co-mitogen to many growth factors, neurofibromin negatively regulates cAMP. Mouse Schwann cells depleted of *NF1* [101] and NF1 MPNST cell lines [116] show increased cAMP levels, which causes aberrant cell proliferation.

Overall, these data suggest that neurofibromin may act upstream of PKA by many ways to regulate cAMP production. On the other hand, it is known that PKA phosphorylates neurofibromin [54] and inhibits its GAP activity by allowing interaction of 14-3-3 with the neurofibromin CTD. It is thus possible that there is a negative feedback mechanism between neurofibromin and PKA. Furthermore, cAMP appears to be a potentially important therapeutic target to correct a set of phenotypes caused by loss or reduced expression of neurofibromin, especially learning defects in NF1-affected children.

### 7.3. Regulation of Dopamine Levels

Neurofibromin functions as a positive regulator of dopamine homeostasis. Indeed, *NF1*-mutant mice display reduced dopamine levels in the brain that lead to deficits in spatial learning, memory, and attention [117,118]. These manifestations were rescued by the administration of L-dopa or methylphenidate, thus supporting the use of methylphenidate to elevate dopamine levels in the treatment of children with NF1-associated learning and attention deficits [119]. Furthermore, a dose-dependent relationship between neurofibromin levels, dopamine signaling, and cognitive deficits was identified in the hippocampus and striatum of NF1 patients [120]. The molecular mechanisms responsible for such regulation are still unknown. Clinical trials conducted with methylphenidate showed an effect in the reduction of attention deficits, spatial working memory impairments, and ADHD symptoms in children with NF1 [121].

Recently, neurofibromin was shown to be involved in the dopamine-mediated production of cAMP in a population of striatal neurons (D2-dopamine receptor expressing medium spiny neurons), suggesting a role in motor learning [122].

### 7.4. Regulation of mTOR Signaling

Neurofibromin also regulates the mTOR pathway. This regulation takes place through the inactivation of the Ras/PI3K pathway, a growth factor-controlled upstream pathway that regulates mTOR (Figure 10).

In the absence of neurofibromin, the hyperactivation of Ras leads to aberrant activation of mTOR via Akt-dependent phosphorylation and inactivation of TSC2, a GAP protein, negatively regulating the small GTPase Rheb. Indeed, mTOR is constitutively active in neurofibromin-deficient primary cells and tumor cell lines derived from NF1 patients. These cells are sensitive to the mTOR inhibitors rapamycin [123,124] and everolimus [125], suggesting that mTOR may constitute a therapeutic target for the treatment of NF1 tumors. Several clinical trials have been developed with different mTOR inhibitors against various manifestations of NF1, sometimes in combination with other drugs. Results were disappointing [126,127,128] except in the case of low-grade pediatric glioma, where a stabilization and a tumor shrinkage was observed [129].

In lysosomes, which constitute a hub for mTORC1 signaling, neurofibromin also regulates mTOR via another nutrient-controlled pathway. Indeed, in an effort to identify new neurofibromin partners using affinity purification followed by mass spectrometry (AP-MS), Li et al. (2017) [130] showed that neurofibromin directly interacts with LAMTOR1, a membrane protein localized to the surface of late endosomes and lysosomes that plays a role in the anchoring of the Ragulator complex (formed by five LAMTOR subunits) to these membranes. This Ragulator complex is crucial for mTORC1 activation on the lysosomal surface in response to amino acids [131]. Functional studies of this interaction demonstrated that neurofibromin negatively regulates mTORC1 signaling in a LAMTOR1-dependent manner [130] (Figure 11).

Thus, neurofibromin negatively regulates the two complementary pathways (controlled by growth factors and amino acids), allowing the activation of mTORC1 in lysosomes.

Xie et al. (2016) [132] described a signaling mechanism in which GPCRs (opioid receptors) activate Ras-AKT-mTOR signaling in the striatum and showed that neurofibromin is required for the cross talk between GPCRs (opioid receptors) and Ras activation. Indeed, upon activation of opioid receptors by morphine, the released Gβγ subunits interact with the SecPH domain of neurofibromin and inhibit its Ras-GAP activity. This results in specific activation of Ras-AKT-mTOR signaling (Figure 12). Deletion of neurofibromin resulted in an increase in Ras baseline activity but abolished the opioid receptor-induced activation of Ras. This mechanism suggests a role for neurofibromin in drug addiction [132].

### 7.5. Control of Actin Cytoskeleton Organization

Actin exists in two forms, monomeric G-actin and polymeric F-actin. The fine-tuning of actin polymerization/depolymerization is important for the regulation of crucial biological processes, such as cell morphology and motility. Cofilin, a member of the ADF family (actin depolymerizing factor), depolymerizes actin by severing aged actin filaments. Phosphorylation of cofilin by LIM kinases (LIMK1 and LIMK2) inactivates its actin-severing activity, resulting in the stabilization of actin filaments, an increase in the number of stress fibers, and focal adhesion formation [133,134].

Aside from its Ras-GAP activity, neurofibromin plays an important role in regulating cytoskeletal organization. Specifically, neurofibromin regulates the dynamic reorganization and turnover of actin filaments through the negative regulation of two parallel pathways: the Rho/ROCK/LIMK2/cofilin and Rac1/Pak1/LIMK1/cofilin pathways. Using *NF1* siRNA, Ozawa et al. (2005) [135] showed that depletion or down-regulation of *NF1* activates the Rho/ROCK/LIMK2/cofilin signaling pathway and leads to the sustained phosphorylation and inactivation of cofilin by LIMK2, which in turn induces the formation of focal adhesions and stable actin stress fibers. The expression of neurofibromin-GRD type 2 (containing exon 23a), but not neurofibromin-GRD type 1, partially restores normal phosphorylated cofilin levels and suppresses the accumulation of actin stress fibers [135]. Consistent with a role of neurofibromin in cell adhesion and motility, Kweh et al. (2009) [61] demonstrated a physical interaction between the neurofibromin CTD and FAK and showed that *NF1*^+/+^ MEF cells adhere less to collagen I and fibronectin I than *NF1*^−/−^ MEF cells and that the two cell genotypes displayed differences in the cellular distribution of actin and FAK [61].

Vallée et al. (2012) [136] described the molecular mechanism involved in the regulation of cytoskeletal dynamics through the Rho/ROCK/LIMK2/cofilin pathway by neurofibromin. Based on a yeast two-hybrid screening approach, they identified an interaction between neurofibromin SecPH and LIMK2. This interaction inhibits LIMK2 kinase activity on cofilin by preventing LIMK2 activation by ROCK, its upstream regulator (Figure 13).

Thus, non-phosphorylated cofilin remains active and able to depolymerize actin, reducing LIMK2-induced actin stress-fiber accumulation [136]. On the other hand, neurofibromin inhibits the parallel signaling pathway centered on LIMK1. Indeed, the Rac1/Pak1/LIMK1/cofilin pathway is inhibited by the pre-GRD of neurofibromin (Nf1_1-1163_) [137] (Figure 13). The pre-GRD was even hypothesized to possess direct Rac-GAP activity. The expression of Nf1_1-1163_ in *NF1*-deficient cells significantly reduced stress-fiber formation and halted cell migration [137].

### 7.6. Microtubule-Dependent Transport in Melanocytes, Neurons, and Schwann Cells

Several independent studies have shown that neurofibromin directly interacts with or belongs to the same protein complex as various proteins involved in the microtubule-dependent transport of organelles, protein complexes, and mRNA in melanocytes, neurons, and Schwann cells. Indeed, in HeLa cells and calf brain, neurofibromin was shown to be in the same complex as kinesin-1, a motor protein involved in anterograde transport along microtubules [138]. De Sheppers et al. (2006) [81] further demonstrated a direct interaction between the neurofibromin GRD and amyloid precursor protein (APP) in melanosomes and APP was previously shown to directly interact with neuronal kinesin-1 and hypothesized to constitute a cargo receptor for kinesin-1 in neurons [139]. Similarly, Arun et al. (2013b) [140] further demonstrated an interaction between the neurofibromin TBD domain and the dynein heavy chain (DHC), a component of the dynein motor protein involved in retrograde transport along microtubules, in melanosomes. Overall, these data strongly suggest that neurofibromin plays an important role in the intracellular transport of melanosomes in melanocytes, which could account for the pathological mechanism of CALM (Café Au Lait Macule) formation. More generally, neurofibromin may play a role in neurotransmitter vesicle trafficking in neurons, which may be involved in cognitive disorders associated with NF1 disease.

Based on the comparison of gene expression in brain regions of WT and *NF1*^+/−^ mice, Donarum et al. (2006) [141] further suggested a functional connection between neurofibromin, APP, and D3R (one of the five dopamine receptors) that interact with APP.

Other data obtained in Schwann cells showed that the neurofibromin TBD is able to interact with LRPPRC (leucine-rich pentatricopeptide repeat motif-containing protein) as part of a ribonucleoprotein complex connected to microtubules via kinesin-1 [142]. These complexes are consistent with RNA granules that contain mRNA and the ribosomal machinery to allow protein synthesis in response to appropriate cues in a temporal and spatial manner [143].

### 7.7. Cell Cycle

Independently of its Ras-GAP function, neurofibromin regulates the metaphase to anaphase transition. Neurofibromin is involved in the spindle assembly checkpoint (SAC) to induce mitotic arrest in response to spindle damage. This function is carried by the CTD, of which the overexpression in yeast delays transition between these two phases of the cell cycle [23].

Koliou et al. (2016) [58] showed that neurofibromin is localized to the mitotic spindle and that such localization may account for a role in regulating chromosome congression during mitosis. These authors further demonstrated that neurofibromin participates in spindle formation and proper chromosome metaphase alignment. Neurofibromin depletion led to aberrant chromosome congression at the metaphase plate, which typically caused chromosomal instability and aneuploidy [58].

### 7.8. Neurofibromin Interactions

The interacting partners of neurofibromin are summarized in Figure 14.

#### 7.8.1. Partners Previously Mentioned in This Review

We have already mentioned that LRPPC and DHC interact with the neurofibromin TBD [140,142], that Ras, Spred1, and APP interact with the neurofibromin GRD [20,76,81], that phospholipids, 5HT6^R^, Gβγ and LIMK2 interact with the neurofibromin SecPH domain [50,113,132,136], that FAK, CASK, and syndecans interact with the neurofibromin CTD [61,62,115], that tubulin interacts with the neurofibromin TBD/GRD and CTD [58,64,65,66], and that kinesin 1 and LAMTOR1 are in the same complex as neurofibromin [130,138].

#### 7.8.2. Other Partners

##### CRMP2

Neurofibromin was found to interact with collapsin response mediator protein-2 (CRMP-2), a protein involved in axonal outgrowth [59,144]. CRMP2 phosphorylation by cyclin-dependent kinase 5 (Cdk5) induces its dissociation from microtubules, leading to collapse of the growth and the arrest of axonal outgrowth. The *NF1*–CRMP2 interaction prevents CRMP2 phosphorylation by Cdk5 and thus promotes neurite outgrowth [59].

Loss of neurofibromin enhances CRMP2 phosphorylation, an event that has been shown to drive glioblastoma cell proliferation and survival. CRMP2 phosphorylation inhibition using (S)-lactosamid reduced glioblastoma cell proliferation, induced apoptosis, and reduced tumor size in a mouse model [145] (Figure 15).

Revealing the neurofibromin-CRMP2 interaction also explained the role of neurofibromin loss in pain suffered by NF1 patients. Indeed, CRMP2 interacts with and positively regulates N-type voltage-gated calcium (CaV2.2) and voltage-gated sodium (NaV1.7) channels, which control sensory neuron excitability. Their deregulation has been shown to be linked to pain syndromes. Increased phosphorylation of CRMP2 following the loss of neurofibromin in sensory neurons enhanced its association with CaV2.2 and NaV1.7, consequently increasing ion channel current densities, leading to hyperalgesia. The inhibition of CRMP2 phosphorylation by (S)-lactosamide normalized channel current densities, excitability, and hyperalgesia [145] (Figure 15).

##### VCP

Neurofibromin was found to interact with VCP (valosin-containing protein) with a domain encompassing residues 1545 to 1950, which contains the SecPH domain [146]. VCP is encoded by the causative gene of IBMPFD (inclusion body myopathy with Paget’s disease of bone and frontotemporal dementia), a dominant inherited disorder with three major symptoms: myopathy, osteolytic bone lesions, and frontotemporal dementia. VCP is an AAA+ ATPase (ATPase associated with diverse cellular activities). It is a multifunctional protein that participates in diverse cellular functions by associating with a large collection of substrates and cofactors that dictate its functionality. Wang et al. (2011) [146] demonstrated that knockdown of VCP decreased dendritic spine density in cultured rat hippocampal neurons, as did NF1 depletion, suggesting that they participate in the same functional pathway. They further obtained data strongly suggesting that alterations in their interaction may contribute to the dementia and cognitive phenotypes observed in IBMPFD and NF1, respectively.

##### Neurofibromin Dimerization

Carnes et al. (2019) [147] were the first to show neurofibromin oligomerisation by studying neurofibromin interactome after affinity purification. Recently, Sherekar et al. (2020) [43] further showed that neurofibromin forms high-affinity dimers in vitro and in cells. Regions responsible for neurofibromin dimerization were mapped between amino acids 1085 and 1172, containing the TBD, and between amino acids 1821 and 2018 (localized after the PH domain and before the CTD). In addition, a mixture of the N-terminal and C-terminal fragments that harbor the dimerization regions can reconstitute full-length dimeric neurofibromin structures with similar high affinity. The biological consequences of neurofibromin dimerization are still unclear [43].

##### Other Partners

Other partners, such as various kinases that phosphorylate neurofibromin (PKA, PKC α and ε), DDHA, involved in neurofibromin phosphorylation, 14-3-3, involved in phosphorylation-dependent interactions, and ubiquitin ligases and UBA (ubiquitin associated) domain-containing proteins, involved in neurofibromin ubiquitination, will be described in the following paragraphs.

This review should not be complete without mentioning keratins regulated by estrogen receptors, even if these interactions remain controversial [147]. The significance of these interactions is unknown but neurofibromin was shown to negatively regulate keratin expression [148].

## 8. Regulation of Neurofibromin Functions by Post-Translational Modifications (PTMs)

### 8.1. Phosphorylation

In 1996, Izawa et al. [54] published the first results on neurofibromin phosphorylation. In this study, they showed that neurofibromin is phosphorylated at its CSRD and CTD by PKA. In addition, they demonstrated the existence of PKA inhibitor-insensitive kinase activity against the CTD, thus demonstrating that PKA is not the unique kinase that phosphorylates the CTD of neurofibromin. At that time, the functional significance of neurofibromin phosphorylation by PKA was unknown [54].

A following study showed that the cellular NO/NOS regulator DDAH interacts with the CSRD and CTD. DDAH was shown to increase PKA-dependent phosphorylation of neurofibromin. Despite its strong interaction with the CTD, the effect of DDAH on neurofibromin phosphorylation was more significant for the CSRD than CTD. The authors suggested that DDAH binding to the CTD may cause an alteration of the tertiary structure of neurofibromin, resulting in an increase of PKA accessibility to the CSRD [56].

In a further study, Feng et al. (2004) [57] provided the first functional insights into neurofibromin phosphorylation. Indeed, they showed that PKA-dependent phosphorylation of neurofibromin on the CTD promotes its interaction with the 14-3-3 protein. Phosphorylation-mediated 14-3-3 binding inhibited the GAP activity of neurofibromin [57].

Neurofibromin is also phosphorylated by the PKC-α isoform on serine residues within the CSRD in response to EGF [55]. PKC-α dependent phosphorylation of the CSRD induces neurofibromin association with the actin cytoskeleton. This results in allosteric regulation of the GRD by increasing its Ras-GAP activity to arrest cell growth (Mangoura et al. 2006). It has also been shown that during neuronal differentiation induced by TPA (a PKC activator), sustained PKC-dependent phosphorylation of neurofibromin is detected at its C-terminus and correlates with prolonged activation of the Ras/ERK pathway [71].

Recently, PKC-ε was shown to be able to phosphorylate neurofibromin on Ser 2808 within the CTD. The phosphorylation of this serine is cell-cycle dependent: it increases during the G2 phase and mitosis and promotes Ran-dependent neurofibromin nuclear accumulation. Ser 2808 is proximal to the functional bipartite NLS (located between amino acids 2555 and 2572) [42]. Therefore, its phosphorylation may regulate NLS accessibility to regulate neurofibromin nuclear entry [58].

### 8.2. Ubiquitination

Ubiquitination consists of a covalent attachment of a small protein called ubiquitin to other protein substrates. This PTM is involved in many biological processes in eukaryotes and regulates protein substrates in various ways, including targeting them for proteasome degradation [149].

In 2003, Cichowski et al. [88] showed that neurofibromin levels are dynamically regulated by the ubiquitin-proteasome pathway. As neurofibromin possesses Ras-GAP activity, its complete degradation is required for maximal Ras activation [88]. In 2009, McGillicuddy et al. [150] showed that protein kinase C (PKC) is activated in response to growth factor stimulation, resulting in a signal that triggers rapid neurofibromin ubiquitination and proteasomal degradation [150]. This process requires sequences adjacent to the N-terminus of the neurofibromin GRD domain [88]. Shortly after its degradation, neurofibromin is re-expressed and its level is re-elevated to downregulate the Ras signal. Thus, PKC regulates the amplitude and duration of Ras-activated signals by promoting neurofibromin degradation.

Phan et al. (2010) [151] used a proteomic approach to identify new binding partners of the Ras-GAP Ira2p in yeast and its mammalian homologue neurofibromin and showed that the proteins containing the UBA domain, Gpb1, and ETEA, negatively regulate Ira2p and neurofibromin, respectively, by promoting their ubiquitination and degradation. ETEA silencing resulted in neurofibromin upregulation, inducing downregulation of Ras-GTP and its downstream effectors ERK and AKT. These results suggest that the inhibition of ETEA may constitute a therapeutic strategy. However, ETEA is not a bona fide E3 ubiquitin ligase.

The first E3 ubiquitin ligase involved in neurofibromin ubiquitination was identified by Tan et al. (2011) [152], who showed that neurofibromin is a substrate for SAG-CUL1-FBXW7 during embryogenesis. SAG-mediated neurofibromin ubiquitination and proteasomal degradation was shown to be required to activate Ras signaling to induce differentiation and proliferation and ensure normal vascular and neural development. Mice deficient in SAG exhibited neurofibromin accumulation, which resulted in the inhibition of Ras signaling and abnormalities in vascular and nervous system development. Based on these results, Tan et al. (2011) [152] proposed SAG E3 ubiquitin ligase as a therapeutic target for NF1 haploinsufficiency.

Gpb1, the protein required for Ira2p ubiquitination and degradation, contains a kelch repeat domain [151]. Interestingly, kelch proteins are known to act as substrate-specific adaptors for Cullin E3 ubiquitin ligase (Cul3) [153]. Consistent with this observation, Hollstein and Cichowski (2013) [154] demonstrated that the ubiquitin ligase complex Cullin3 and its adaptor protein, kelch repeat and BTB domain-containing 7 (KBTBD7), catalyzes PKC-mediated neurofibromin ubiquitination. In addition to genetic inactivation of *NF1*, tumorigenesis can be promoted by neurofibromin destabilization, resulting from excessive proteasomal degradation mediated by hyperactivation of PKC under pathological conditions. Based on this observation, glioblastoma, in which neurofibromin is destabilized, was shown to be sensitive to PKC inhibitors. Cul3 inhibition also stabilized neurofibromin when the protein was destabilized and the *NF1* gene was intact. Thus, PKC and Cul3 were suggested to be potential therapeutic targets for the treatment of gliomas that exhibit destabilized neurofibromin and the wild-type *NF1* gene [150,154]

Recently, Green et al. (2019) [155] reported that hypoxia-associated factor (HAF) promotes neurofibromin ubiquitination and proteasomal degradation, especially under hypoxic conditions. This resulted in activation of the Ras-ERK pathway [155] (Figure 16).

### 8.3. SUMOylation

We previously showed that a fraction of neurofibromin colocalizes with PML-NBs in the nucleus of the astrocytoma cell line: CCF [72]. PML-NBs are dynamic protein-containing structures, for which PML protein is their key organizer. PML protein localizes at their surface to recruit an ever-growing number of proteins, of which their most commonly known features are SUMOylation or the presence of SUMO interaction motifs (SIM) [156,157,158]. Neurofibromin contains 15 SUMO consensus motifs and two SIM predicted by JASSA [159] and SUMOPLOT (https://www.abgent.com/sumoplot). These observations suggest that neurofibromin may be sumoylated. In our previous study, our experiments to demonstrate neurofibromin sumoylation were inconclusive. Since then, neurofibromin has been identified in two systematic studies of the sumoylated proteome [160,161]. The latter study identified two sumoylation sites in exon 23a at K1383 and 1385. However, no specific study on neurofibromin sumoylation has been undertaken. We performed new experiments and recently succeeded in showing that neurofibromin is indeed a SUMO target and carried out a functional study of neurofibromin sumoylation (manuscript in preparation).

## 9. Conclusions

In conclusion, neurofibromin is a multifunctional protein that impacts several crucial cellular processes in cells of different tissues, including proliferation, growth, division, survival, and migration. It guides these processes by directly acting on various signaling pathways. Sometimes, these pathways converge to a single cellular process, thus allowing neurofibromin to intervene at different levels of the same process. Each neurofibromin action is tightly regulated by a wide range of PTM and interactions but also by the differential expression of neurofibromin isoforms with different cellular locations and functions. The multifunctionality of neurofibromin and the complexity of its regulation, along with its ubiquitous expression, explain the diversity of phenotypes observed and the number of tissues affected in neurofibromin patients.

Before proposing appropriate therapeutic strategies against neurofibromin symptoms, it was essential to first determine the functions of neurofibromin and the molecular mechanisms of their regulation. Over the last 30 years, many groups have made remarkable efforts in this endeavor. However, despite the considerable amount of data generated on the biochemistry of neurofibromin (structure, functions, interactions, localization, and regulation by PTMs) leading to the identification of several therapeutic targets and recent approval of the first therapy (selumetinib) for inoperable plexiform neurofibroma associated with NF1 (by targeting MEK kinase) [162], we are still far from having a complete picture of the functions of *NF1* and their regulation. Indeed, the crystallographic structure has been resolved for only the two central domains, GRD and SecPH, which constitute no more than 20% of the entire protein. Although many of the functions of neurofibromin are mediated by these two domains; the pre-GRD and CTD also play crucial roles in the functions of neurofibromin. Indeed, there is no hot spot of mutations that leads to NF1 disease and the identified NF1 patient mutations are distributed throughout the *NF1* gene, suggesting that all neurofibromin domains have important functions yet to be discovered or further characterized. One of the challenging technical issues of research on neurofibromin has been the unavailability of constructs containing the entire *NF1* cDNA because of its size, its high mutational rate, and its toxicity for *E coli*. Recently, different solutions have been proposed to circumvent this problem. Wallis et al. (2018) [163] developed and validated a mouse *NF1* cDNA expression system, Cui and Morrison (2019) [164] published experimental conditions to clone the full length *NF1* containing a mini-intron to eliminate its toxicity, and Sherekar et al. (2020) [43] developed a codon-optimized human *NF1* cDNA that is stable and non-toxic in *E. coli* and human cells. These constructs will be of great help in studying the entire neurofibromin protein instead of separate domains. They will make it possible to examine the biochemical effects of any *NF1* genetic variant and have an integrated view of their impact on neurofibromin functions at the molecular and cellular level.

## Figures and Tables

**Figure 1 cells-09-02365-f001:**
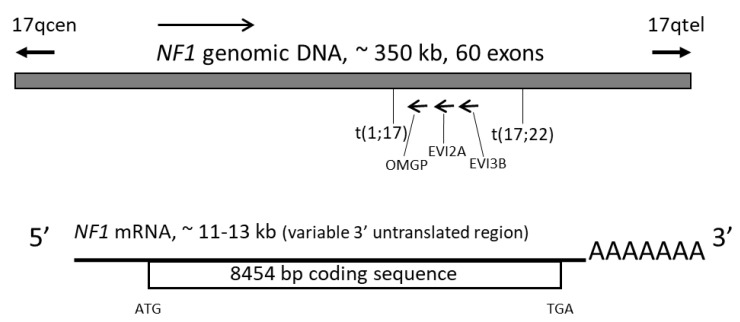
Schematic representation of the *NF1* gene and its mRNA transcript; kb: kilo bases

**Figure 2 cells-09-02365-f002:**
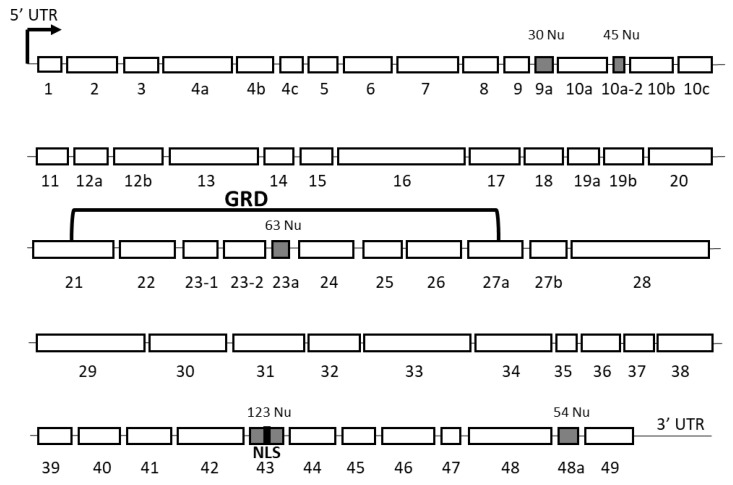
Schematic representation of the exons of the full-length transcript of *NF1* according to the old nomenclature. The alternatively spliced exons are indicated in grey with their number of nucleotides. The nuclear localization signal (NLS) in exon 43 (in black) and the GAP-related domain (GRD) are indicated; Nu: nucleotides.

**Figure 3 cells-09-02365-f003:**
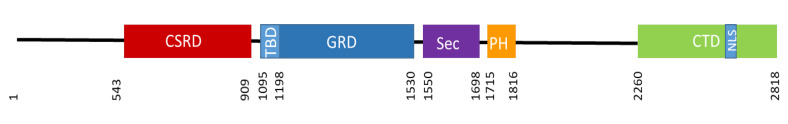
Schematic representation of neurofibromin domains. CSRD (cysteine- and serine-rich domain) in red, TBD (tubulin-binding domain) in light blue, GRD (GAP-related domain) in blue, Sec (Sec14 homologous domain) in purple, PH (pleckstrin homologous domain) in orange, CTD (C-terminal domain) in green, NLS (nuclear localization signal) in blue. The amino acid number is indicated below.

**Figure 4 cells-09-02365-f004:**
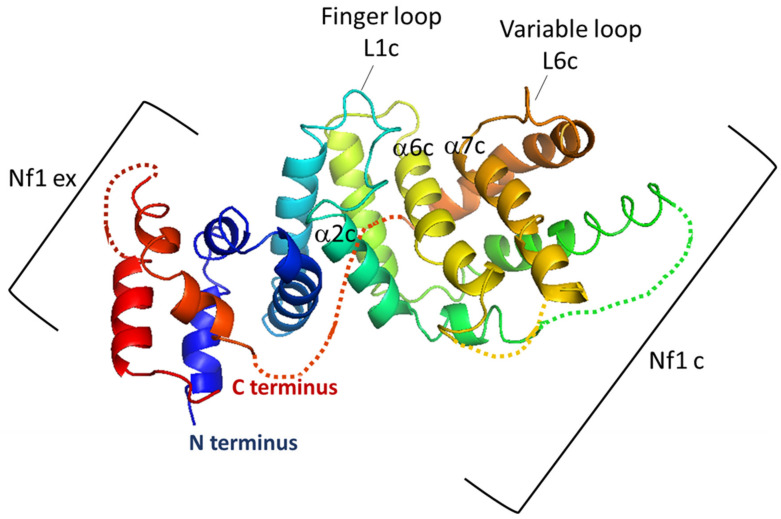
Structure of neurofibromin-GRD domain (NF1-333) in a ribbon representation. The central domain (NF1c) is shown in light blue, green, yellow, and brown and the extra domain (NF1ex) in dark blue and red. Regions that are not visible in the neurofibromin-GRD model were complemented by the corresponding segments derived from the GAP-334 model and are shown as dotted lines. Helices α6c and α7c, forming the bottom of the Ras-binding groove, are indicated. The variable loop (L6c) and α2c helix, involved in the interaction with Ras, and finger loop (L1c), which provides an Arg residue (R1276) to the active site of Ras to stabilize the transition state of the GTPase reaction, are also indicated.

**Figure 5 cells-09-02365-f005:**
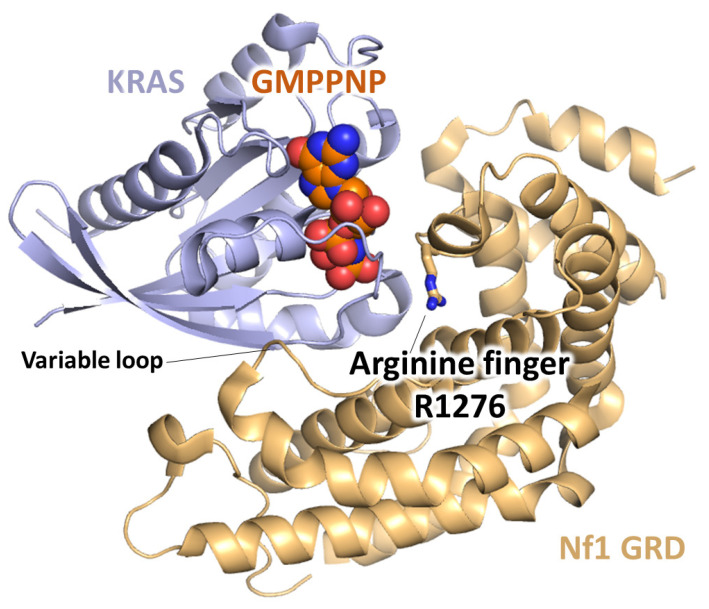
Structure of a KRas-neurofibromin GRD complex in a ribbon representation. The arginine finger (R1276) and variable loop are shown.

**Figure 6 cells-09-02365-f006:**
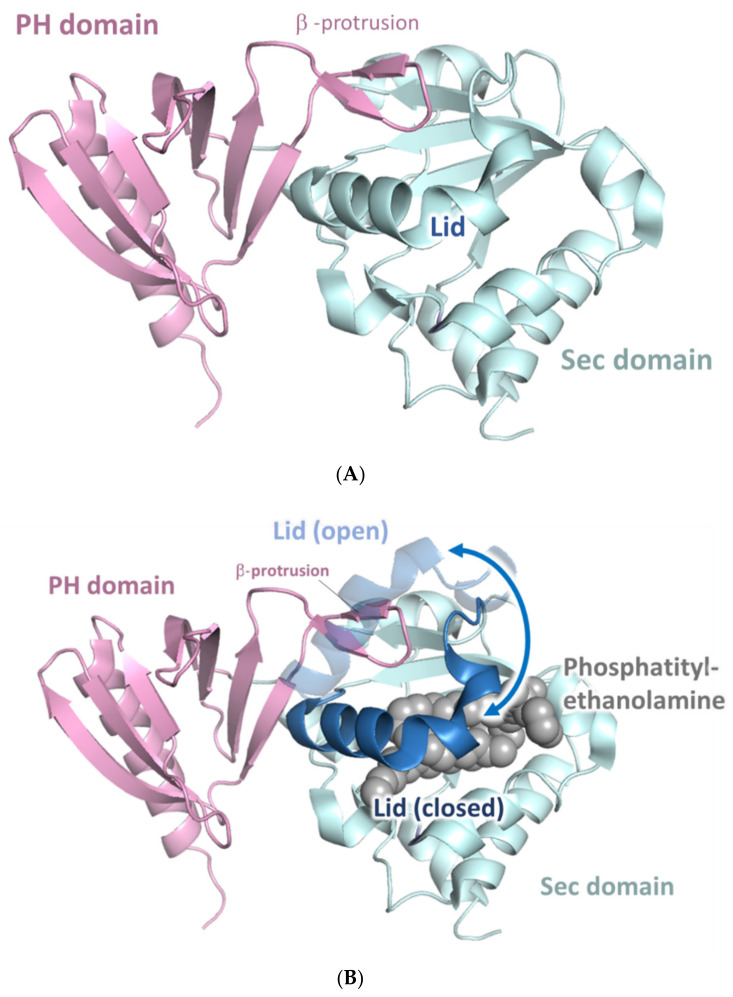
Structure of the SecPH domain: (**A**) Ribbon representation of the SecPH domain of human neurofibromin [50]. (**B**) Ribbon representation of neurofibromin-SecPH superimposed over the open Sec conformation (derived from the structure of Sec14p) [51].

**Figure 7 cells-09-02365-f007:**
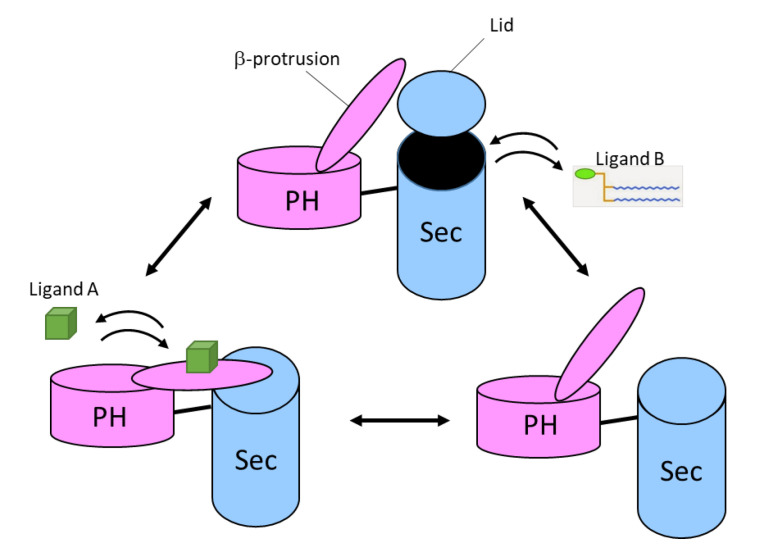
Proposed functional mechanism of the neurofibromin-SecPH module. Hypothetical mechanism of how conformational changes in the neurofibromin-PH domain upon binding of ligand A may regulate access to ligand B for the neurofibromin-Sec lipid-binding cage.

**Figure 8 cells-09-02365-f008:**
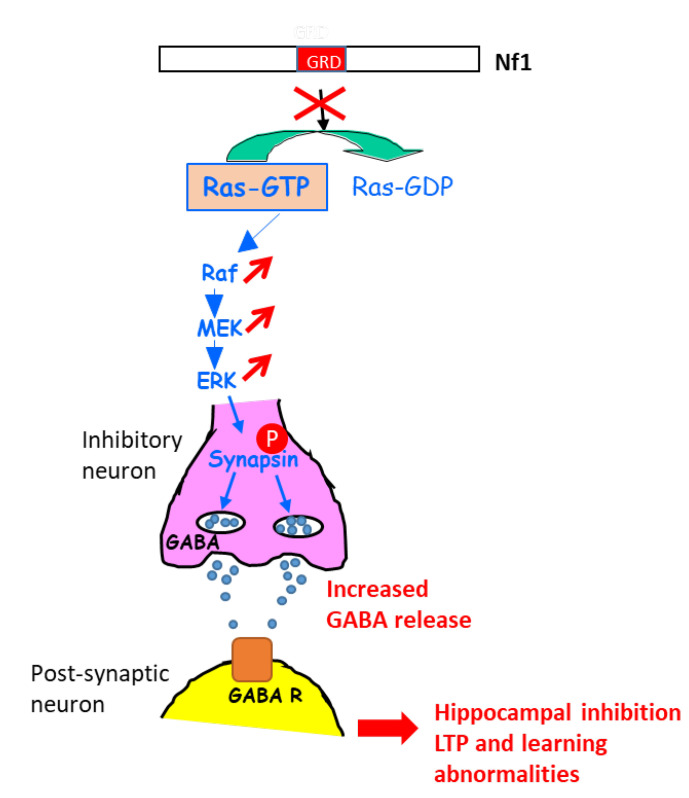
Schematic representation explaining the increase in GABA secretion upon the activation of ERK in *NF1*-deficient inhibitory neurons. GABA R: GABA receptor.

**Figure 9 cells-09-02365-f009:**
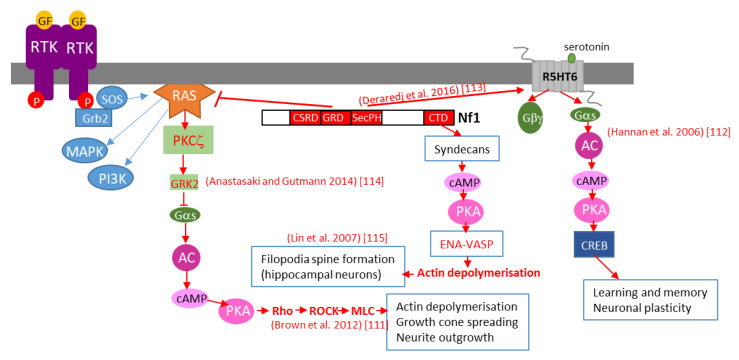
Schematic representation of various mechanisms of neurofibromin-mediated cAMP regulation. GF: growth factor, RTK: tyrosine kinase receptor, AC: adenylate cyclase, CREB: cAMP response element-binding protein, MAPK: mitogen-activated protein kinase, PI3K: phosphoinositide 3-kinase, SOS: son of sevenless, Grb2: growth factor receptor-bound protein 2.

**Figure 10 cells-09-02365-f010:**
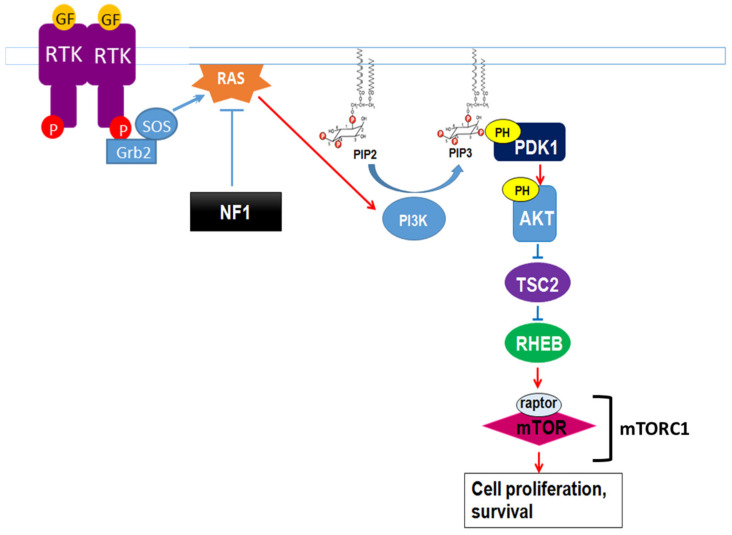
Neurofibromin regulation of the growth factor-controlled Ras/PI3K signaling pathway. PIP2: phosphatidylinositol-4,5-bisphosphate, PIP3: phosphatidylinositol 3,4,5 trisphosphate, PDK1: phosphoinositide-dependent kinase-1, PH: pleckstrin-homologous domain, TSC2: TSC complex subunit 2, RHEB: Ras homolog enriched in brain.

**Figure 11 cells-09-02365-f011:**
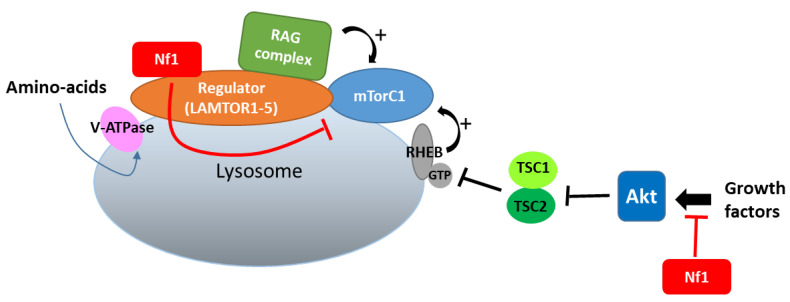
Neurofibromin binds to LAMTOR1 and inhibits mTORC1 signaling.

**Figure 12 cells-09-02365-f012:**
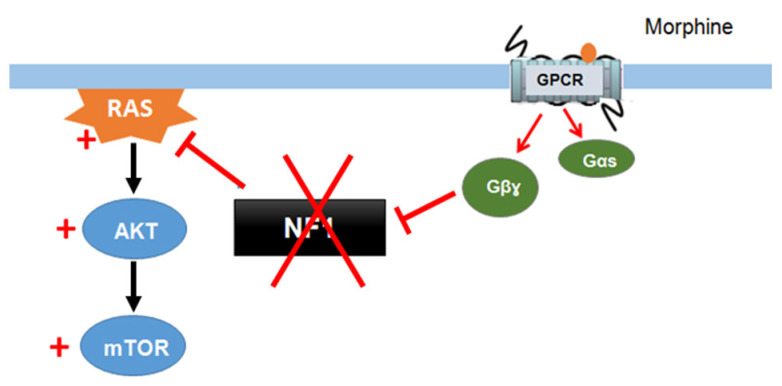
Neurofibromin mediates cross talk between GPCR (opioid receptors) and Ras/AKT signaling [132].

**Figure 13 cells-09-02365-f013:**
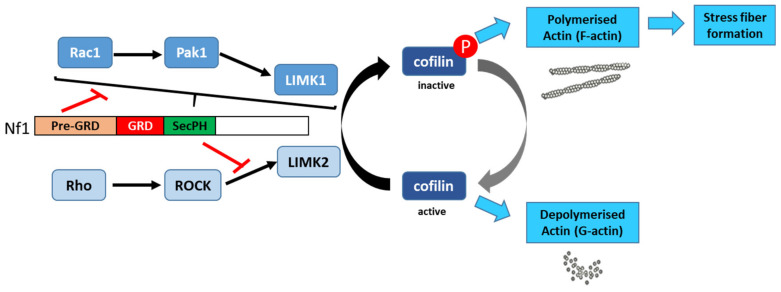
Schematic representation of the mechanisms for neurofibromin-mediated regulation of the actin cytoskeleton [136,137].

**Figure 14 cells-09-02365-f014:**
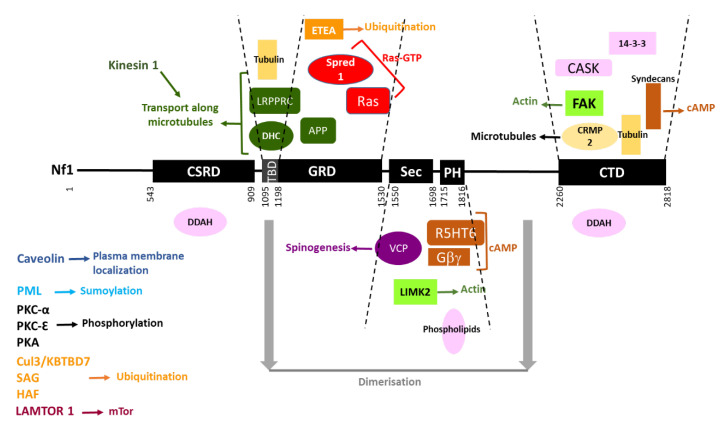
Neurofibromin interacting partners. Neurofibromin domains are indicated as black boxes and their limits are written below. Interacting partners of each domain are pictured between the dashed lines delimiting the domains. Each type of function is represented by a color: Ras-GTP in red, cAMP in brown, actin dynamics in green, microtubules in yellow, transport along microtubules in dark green, spinogenesis in purple, ubiquitination in orange, sumoylation in blue, phosphorylation in black, plasma membrane localization in dark blue, and unknown function in pink. Regions necessary for neurofibromin dimerization are indicated in grey. Partners interacting with unknown domains of neurofibromin or being part of the same complex are not pictured but their name is indicated on the left side of the figure.

**Figure 15 cells-09-02365-f015:**
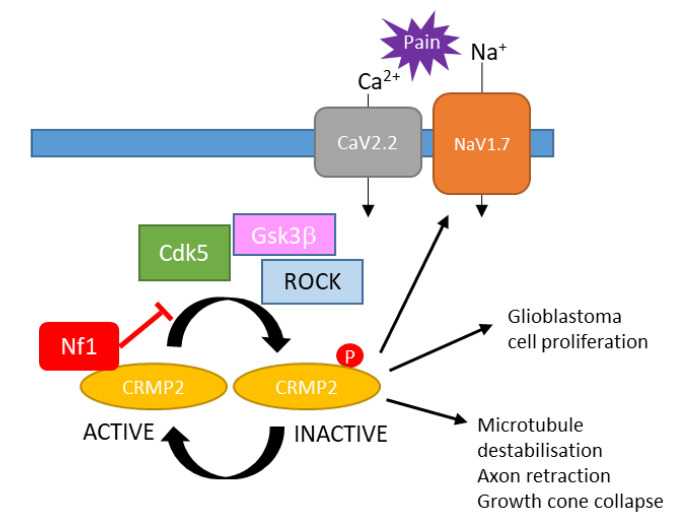
Schematic representation of the mechanisms for neurofibromin-mediated inhibition of CRMP2 phosphorylation and inactivation [59,145].

**Figure 16 cells-09-02365-f016:**
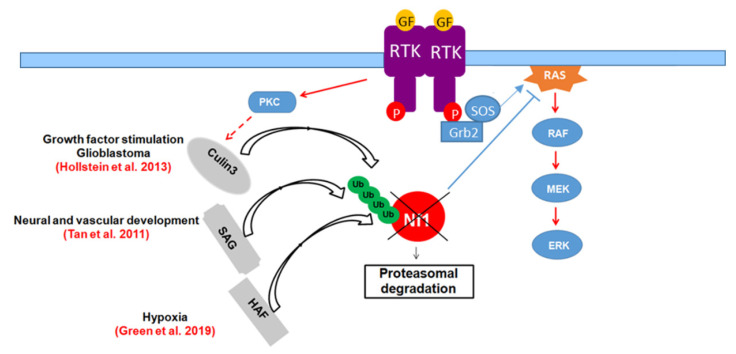
Regulation of neurofibromin stability by the ubiquitin-proteasome pathway.

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
