# Peer review of "Neurofibromin Structure, Functions and Regulation"

_cells, 2020, doi:10.3390/cells9112365_

Round 1
Reviewer 1 Report
Bergoug et al. have written a nice review summarizing much of the data that exists regarding neurofibromin structure, function, interactions, and mechanisms of regulation. There are several points that could improve the manuscript.
- Please stick to the general nomenclature for gene names:
- NF1 without italics refers to the syndrome
- NF1 in italics in human when referring to the gene
- Nf1 in italics in mouse when referring to the gene
- neurofibromin (rather than saying Nf1) when referring to the gene would avoid confusion
- I would discuss PMID 31836666 Sherekar et al. JBC 2020 in the structure section. When reading, I wondered why this paper was not discussed and then found it later in the protein interaction section. I think it should be discussed first in the structure section and then mentioned briefly in the later section.
- Most of the references in this paper are older references. In some areas in which there are implications made that there may be clinical relevance to a particular pre-clinical finding based on older literature, I think it is important to point out the more recently published clinical trials in that area that have failed pointing toward the need for further research and conversely point out the situations in which we have been successful.
-Line 372. Farnesyltransferase inhibitors failed in clinical trials. PMID 24500418 Widemann et al. 2014
-Line 380 you reference numerous preclinical studies in which lovastatin demonstrated benefits in terms of learning/cognition. However clinical trials have not panned out. PMID27956565 Payne et al. Neurology 2016
-Line 463 there is some clinical data to suggest that altering dopamine levels is beneficial for ADHD in the setting of NF1. PMID 30166301 Pride et al. BMS Open 2018.
-Line 483. There are numerous trials to reference that have not panned out for mtor inhibitors in NF1 associated tumors in humans. Even the combinations tested thus far have all been negative trials. This needs to be noted.
- As mentioned earlier, the discussion of neurofibromin dimerization would be best discussed in the section on structure
Author Response
We would like to thank you for your positive comments on our review and for your very pertinent suggestions which will greatly improve it. We have followed all your suggestions.
-First, Nf1 has been replaced everywhere in the text by neurofibromin and we have corrected the mistakes in nomenclature for the gene names (human and mouse).
-Sherekar et al., 2020 has been discussed in section 5. Related to neurofibromin structure and domains. A specific paragraph, 5.4, entitled « Structural data on full-length neurofibromin » has been added at the end of the section.
-Everytime preclinical data on a particular therapy depicted in our manuscript has been followed by clinical trials, we have mentionned it and indicated the results obtained and the reference. This is the case line 406 for farnesyltransferase inhibitors, line 415 for lovastatin and simvastatin, line 508 for methylphenidate, line 527 for mTOR inhibitors.
Reviewer 2 Report
This is an excellent and comprehensive review of the neurofibromin protein and it's biochemical and structural understanding. Overall, it is very well written and extremely thorough and the field has been in need of such a review for a long time. I have a few minor comments which might improve the manuscript slightly:
Section 4.1: The authors need to explain in section 4.1 the complexity of the nomenclature around NF1 transcripts. The literature is very confusing on this topic, as the NCBI Genbank RefSeq lists the shorter 2818 aa sequence as isoform 2 (NM_000267.3), and the longer 2839 aa sequence as isoform 1 (NM_001042492.3). However, many protein database (including UniProt) label them the other way. This has led to significant confusion in the field, and it would be good to clarify this more in this review, perhaps citing the RefSeq numbering as well.
Section 4.1.3: please add a reference for the claim in paragraph 1 of this section (lines 121-123)
Section 4: perhaps worth adding a paragraph to discuss homology of NF1 in other organisms? You mention yeast, but the conservation of NF1 throughout mammals and insects is quite impressive and maybe worth a brief discussion?
Section 9: While the Cui and Morrison paper offers one solution to stabilizing NF1 clones, many groups have already been working with optimized versions of the sequence which are highly stable in E. coli and don’t require the mini-intron strategy. For instance, the construct described in Sherekar et al which was derived from Addgene #70423). So there are multiple ways that one could stabilize NF1 and these constructs are widely used in the field these days.
One manuscript which probably should also be cited is Carnes et al (Genes 2019, 10, 650) which looks at interactions of Neurofibromin with keratins. While somewhat controversial still, it may be a good addition to the section on binding interactors.
Author Response
We would like to thank you for your complimentary comments on our review and for all your remarks which really point to gaps or areas of confusion that needed to be addressed. We have followed all your suggestions to improve our review.
-First, we explain in section 4.1 (line 100) the complexity of the nomenclature around NF1 transcripts. We have cited RefSeq numbering to avoid any confusion.
-A new paragraph has been inserted in section 3., page 3 line 83, about conservation of NF1 in different species.
-Section 9, attempts to entire NF1 clones stabilization are now more fully described (line 811).
-Keratins interactors are now mentionned and the reference suggested is cited